**Increased inorganic aerosol fraction contributes to air pollution and**
**haze in China**

Yonghong Wang[1,2], Yuesi Wang [1,6,8], Lili Wang[1], Tuukka Petäjä[2,3], Qiaozhi

Zha[2],Chongshui Gong[1,4],Sixuan Li[7], Yuepeng Pan[1], Bo Hu[1], Jinyuan Xin[1] and

Markku Kulmala[2,3,5]

[1]State Key Laboratory of Atmospheric Boundary Layer Physics and Atmospheric Chemistry
(LAPC), Institute of Atmospheric Physics, Chinese Academy of Sciences, Beijing 100029,
China
[2]Institute for Atmospheric and Earth System Research / Physics, Faculty of Science, P.O.Box
64, 00014 University of Helsinki, Helsinki, Finland
[3]Joint international research Laboratory of Atmospheric and Earth SysTem sciences
(JirLATEST), Nanjing University, Nanjing, China
[4]Institute of Arid meteorology, China Meteorological Administration, Lanzhou 730000,
China
[5]Aerosol and Haze Laboratory, Beijing Advanced Innovation Center for Soft Matter Science
and Engineering, Beijing University of Chemical Technology (BUCT), Beijing, China
[6]Centre for Excellence  in Atmospheric Urban Environment, Institute of Urban Environment,
Chinese Academy of Science, Xiamen, Fujian 361021,China
[7]State Key Laboratory of Numerical Modeling for Atmospheric Sciences and Geophysical
Fluid Dynamics (LASG), Institute of Atmospheric Physics, Chinese Academy of Sciences,
Beijing 100029, China
[8]University of Chinese Academy of Sciences, Beijing 100049, China
Revised to: Atmospheric Chemistry and Physics
Corresponding authors: Y.S. Wang, L.L.Wang and M. Kulmala
E-mail: wys@mail.iap.ac.cn;wll@mail.iap.ac.cn; markku.kulmala@helsinki.fi

## Abstract

The detailed formation mechanism of increased number of haze events in China is
still not very clear. Here, we found that reduced surface visibility from 1980-2010 and
an increase in satellite derived columnar concentrations of inorganic precursor from
2002 to 2012 are connected with each other. Typically higher inorganic mass fractions
lead to increased aerosol water uptake and light scattering ability in elevated relative
humidity. Satellite observation of aerosol precursors of $NO_2$ and $SO_2$ showed increased
concentrations during study period. Our in-situ measurement of aerosol chemical
composition in Beijing also confirmed increased contribution of inorganic aerosol
fraction as a function of increased particle pollution level. Our investigations
demonstrate that the increased inorganic fraction in the aerosol particles is a key
component in the frequently occurring haze days during studying period, and
particularly the reduction of nitrate, sulfate and their precursor gases would contribute
towards better visibility in China.

## Introduction

As one of the most heavily polluted regions in the world, China has suffered from
air pollution for decades (Hao et al., 2007; Zhang et al., 2015). Aerosol particles, as
major  air pollutant, have significant effects on human health (Lelieveld et al., 2015).
The general public and the central government of China have realized the severe
situation and have taken some actions to improve the air quality nationwide in the recent
years. For example, the state council published a plan for air pollution control, in

September of 2013, aim to reduce $PM_{2.5}$ concentrations by 10%~25% in different regions of China. The successful implementation requires a sufficient knowledge of haze formation mechanism (Kulmala, 2015) and comprehensive observation network (Kulmala, 2018). Our understanding on haze events with high $PM_{2.5}$ concentrations in China is still limited due to the spatial-temporal variation of aerosol properties and limited observation information (Wang et al., 2016). Recent studies found that secondary aerosol components are important during the intense haze events in Beijing, Xi'an, Chengdu and Guangzhou during January of 2013, and the reduction of aerosol precursors is a key step to reduce particle pollution (Guo et al., 2014; Huang et al., 2014). The analysis of longer time series data from Nanjing shows that secondary particles are typically dominating even the number concentrations in polluted conditions (Kulmala ., 2016). A recent study have suggested significantly decreased trends of $PM_{2.5}$ and $SO_2$ in China from 2015-2017 by analyzing data sets from Ministry of Ecology and Environment of China (Silver et al., 2018). The column $NO_2$ concentration obtained from OMI showed increased trend during 2005-2011, while a decreasing trend during 2012-2015 (Itahashi et al., 2016). The $SO_2$ concentration has decreased around 50% from 2012-2015 in North China Plain due to economic slowdown and governments efforts to restrain emissions from power and industrial sectors (Krotkov et al., 2016). However, the most abundant mass fractions of atmospheric aerosol are inorganic and organic components, which have large spatio-temporal variation (Jimenez et al., 2009). Identifying the most abundant as well as critical aerosol species that contribute to the haze formation in a longtime perspective is important to draw up effective plans for the air pollution control.

Here, a comprehensive data sets were used to reveal that an increasing trend of inorganic components in atmospheric aerosol may be a pivotal factor, at least, which

leads to frequently occurred haze events in China from 1980-2010. We suggests that
the controlling of inorganic aerosol components of nitrate, sulfate and their precursors
should be of a high priority due to their strong water uptake abilities and therefore, light
scattering ability in high RH conditions.

## 73 **2.Methodology**

The daily averaged visibility and relative humidity data in 262 sites of China are
obtained from the Integrated Surface Dataset (ISD) from National Oceanic and
Atmospheric Administration National Climate Data Center of the USA from 1980-
2010 (https://www.ncdc.noaa.gov/isd). The visibility observations were made three
times a day at 8-hour intervals begins at 00:00 by well trained technicians. They
measured visual range using distinctive markers, such as tall buildings, mountains and
towers, to which the distance from the meteorological monitoring stations are known.
We quantified the importance of relative humidity to visibility as the hygroscopic
inorganic compounds typically grow in size in high humidity (Swietlicki et al., 2008).
Aerosol size growth and composition change in high humidity condition are highly
related light scattering ability (Zhang et al., 2015). Studies always use $f$ (RH), a
parameter which is defined as the ratio of light scattering coefficient under high RH
with that under low RH. $f$ (RH) is a unitless number, usually ranges from one to two.
At ambient RH around 80%, a higher $f$ (RH) value usually corespondes to higher
inorganic aerosol fraction, while a lower value usually corespondes to high organic
fraction. The reason is that inorganic aerosol compounds of nitrate,sulfate and
ammonium have more strong water uptake ability than organic comopounds. In
addition, the high humidity condition in ambient prefers the formation of inroganici
aerosol from precursors of $NO_2$ and $SO_2$ (Wang et al., 2014). In this study, for a given
site and given year, we defined a $f$ (RH)-like parameter, Ri, using the observed annual
visibility (V) as a ratio ($R_i$) between visibility values from the surface observation
stations, when the daily average RH was below 40% for more than 20 days. In the
corresponding high-humidity cases daily RH was between 80%~90% for more than
20 days each year at a given observation site:

$R_i = \frac{V_{dry}}{V_{wet}}.$
We use this ratio to infer long trend of aerosol hygroscopicity information. In
addition, we calculate anomaly (A) from the ratio for a given year $i$ as a difference from
the 30-year ($R_{30y}$) from 1980 to 2010:
$$A = R_i - R_{30y}.$$
Our spatial focus is placed on North China Plain, Yangtze River Plain and Sichuan
Basin due to frequent haze events (Zhang et al., 2012). The  stations in Pearl River delta
region and other Southern China stations were not included due to limited days with
the daily average RH below 40%.
The atmospheric column amount of $NO_2$ and $SO_2$ data are obtained from 2002-2012
and 2004-2012, respectively, from SCIAMACHY (Scanning Imaging Absorption
spectrometer for Atmospheric CHartographY) satellite products. SCIAMACHY is an
atmospheric sensor aboard the European satellite ENVISAT. It was launched in March 2002 as
a joint project of Germany, the Netherlands and Belgium. It measures atmospheric absorption
in spectral bands from the ultraviolet to the near infrared (240 nm - 2380 nm), allows to retrieve
atmospheric column concentrations of $O_3$, BrO, OClO, ClO, $SO_2$, $H_2CO$, NO, $NO_2$, $NO_3$, CO,
$CO_2$, $CH_4$, $H_2O$, $N_2O$, aerosols, radiation and cloud properties (Boersma et al., 2004) . Aerosol

chemical composition from GEOS (Goddard Earth Observing System)-Chem chemical transport model combined with satellite AOD products in China during 1998-2012 is used. The model utilizes assimilated meteorology data and regional emission inventories with a horizontal resolution of 2°×2° with 47 vertical levels from surface to 80 km. The $PM_{2.5}$ concentration was retrieved from AOD of satellite and the relationship between $PM_{2.5}$ and AOD in GEOS-Chem. The detailed information about the model can be found in (Boys et al., 2014). Aerosol chemical composition of organic, sulfate, nitrate, ammonium and chloride were measured with a high-resolution-time of flight-aerosol mass spectrometers during an intensive campaign in urban Beijing from November of 2010 to January of 2011 (DeCarlo et al., 2006). Detailed information of data analysis, collection efficiencies (CE) and relative ionization efficiencies are presented in Zhang et al. (2014).

## 3. Results and discussion

### 3.1 Decreasing trend in visibility in high relative humidity conditions

According to the geographical division, our study sites are mainly in North China Plain (NCP), Sichuan Basin (SCB) and Yangtze River Plain (YRP) as showed in Figure 1. The average visibility in low RH in NCP, SCB, YRP and China are 18.2 km, 21.4 km, 19.5 km and 23.3 km, while the values in high RH conditions are 10.6 km, 13.7 km, 13.7 km and 17.4 km, respectively. In general, visibility in low RH condition has fluctuated trend, particularly in Northern China Plain, Sichuan Basin and Yangtze river Plain region, whereas visibility in high RH conditions showed decreasing trend as shown in Figure S1 (a) and (b) . The average ratio of visibility in low RH to that in high RH from 1980-2010 is presented in Figure 1. The maximum ratios were identified in eastern China and in some western Chinese cities. Three heavily polluted regions,

Northern China Plain, Sichuan Basin and Yangtze river Plain were identified based on
values of high $R_i$, which are also constant with aerosol mass concentrations and haze
day distributions (van Donkelaar et al., 2010; Xin et al., 2015). That is, the higher ratios
occurred in more severe air pollution areas, like North China Plain, Sichuan Basin and
the city of Urumqi, where the contribution of hygroscopic aerosol is more pronounced
in comparison with non-hygroscopic dust particles. The average $R_i$ during 1980-1984
in Northern China Plain, Sichuan Basin and Yangtze river Plain are 1.62, 1.41, 1.29
and 1.31, respectively, contrasting with the values of 1.98, 1.81, 1.70 and 1.52 during
2006-2010. The increments are 22.3%, 27.3%, 31% and 16%, respectively. It is worth
noting that the $R_i$ in Yangtze river Plain region exhibits the most increment, which
implies the increased emissions with rapid economic growth. Long time trends of this
ratio in a specific site can reveal the variation of aerosol inorganic fraction and organic
fraction due to their different hygroscopicity and water uptake capacity and associated
light extinction ability. That is, the mass fractions and concentrations of sulfate, nitrate
and ammonium may have increased over study period as they dominate water uptake
ability compared with other components (e.g., organic, black carbon, dust and metal
elements, see Table S1) in the atmospheric aerosol (Wang et al., 2015). For the selected
regions, we have calculated the anomaly as a regional average as shown in Figure 2.
The ratio showed increasing trends over three regions of China and the maximum trends
occurred in North China Plain with the value of 0.0168 per year, which indicate an
increase of hygroscopic aerosol in these regions during the 30-year observation period.
To corroborate our results, Yang et al. (2011) showed an increasing fraction of
inorganic components by 20% in Beijing from 1998 to 2008 using in-situ offline aerosol
chemical composition measurement, especially in summer, while the fractions of
hydrophobic components such as organic and black carbon decreased in the aerosol
phase. A study by Boys et al. (2014) revealed  that increasing fraction of secondary
inorganic aerosol is dominated in the increased mass concentration of $PM_{2.5}$ in China
from 1998-2012 using GEOS-Chem model combined satellite results. By using
observed meteorology data sets,  Fu et al. (2014) revealed that the number of haze days
have significantly increased in the past three decades over North China Plain due to the
increase in hygroscopic inorganic aerosol composition.
**3.2 Enhanced emissions of inorganic aerosol precursors**
The longterm trends of aerosol precursors and their spatial variability can improve
our understanding of the trends in aerosol chemical composition. Figure 3 and Figure
4 show atmospheric column trends of $NO_2$ and $SO_2$ observed from SCIAMACHY. The
column $NO_2$ level can be a good proxy for vehicle and coal burning emission associated
with oil and coal consumption (Richter et al., 2005). The column amount of  $NO_2$
showed pronounced increasing trends in three regions of China, particularly in Northern
China with the value of $0.14 \times 10^{15}$ molecule/cm$^2$/year from 2002 to 2011. This is
probably associated with the increase in power plant and on-road vehicle emissions
(Wu et al., 2012; Krotkov et al., 2016). The average $NO_2$ concentration in Northern
China increased by more than two-fold, while in the Yangze River Plain region
experienced a considerable smaller trend in $NO_2$, with the value of $9.7 \times 10^{15}$
molecule/cm$^2$ in  2010 and $6.4 \times 10^{15}$ molecule/cm$^2$ in 2002. It is worting noting a
dectresed trend occurred during year 2008, which is mainly due to emission  reduction
during the Olympic games and economic downturn (Lin and McElroy, 2011). As a
whole, the column $NO_2$ concentration in China doubled from 2002 to 2010, with the
values of $1.4 \times 10^{15}$ molecule/cm$^2$ in 2002 and $2.8 \times 10^{15}$ molecule/cm$^2$ in 2010,
respectively.

Figure 4 depicts trend in $SO_2$ concentration in four regions of China from 2004 to

2010. The $SO_2$ concentration showed an increasing trends in North China Plain,
Sichuan Basin and Yangze River Plain, but increased mostly in China from 2004 to
2012. A decreasing trend was observed during the year of 2008 and 2009, especially in
Northern China Plain. This may be due to a combination of Chinese economic
downturn and emission reduction during the Olympic games (Lin and McElroy, 2011)
(Wang et al., 2010). Anyway, as an important aerosol precursor, $NO_2$ showed the
most increasing trend in China from 2002-2012 , consistent with the trend of increased
aerosol concentration by modeling result (Xing et al., 2015). Figure S3 shows the
annual trends of aerosol inorganic fraction in $PM_{2.5}$ mass concentration from 1998-2012
with GEOS-Chem model combined with satellite results in China. The results indicate
that North China Plain area suffered the most from heavily pollution, consistent with
our surface observations (Xin et al., 2015). Aerosol concentrations have increased
considereably from 1980 to 2010. The modeling combined with satellite results by Boys et al.
(2014) show that concurrently the fraction of inorganic fraction has increased more rapidly.
Consequently, the water uptake of the aerosol have increased leading to reduced visibility as
we suggested, which is consistent with ground-based observations (Yang et al., 2011).
**3.3 Validation of increased inorganic aerosol components with elevated air**
**pollution level with in-situ measurement**

To validate our hypothesis that the increased inorganic components contribute to

visibility degradation, we used four months of aerosol chemical composition and
visibility data from urban Beijing from November of 2010 to February of 2011. As
shown in Figure 5, we divided the visibility values into four bins, which corresponds to
clean time to heavy pollution time and to conditions in between. The inorganic aerosol
precursors of $SO_2$ and $NO_2$ nearly doubled as the visibility decreased from more than
10 km (clean time) to less than 2 km (heavily polluted time). At the same time, the mass
concentrations of nitrate, sulfate and ammonium components increased to 13.5 $\mu g\ m^{-3}$,
15.5 $\mu g\ m^{-3}$ and 10.6 $\mu g\ m^{-3}$, respectively. Meanwhile, the mass fraction of these
inorganics increased from 11.3% to 17.3%, from 13.0% to 19.9% and from 9.6% to
13.6%, respectively. At the same time, the mass concentration and fraction of organic
components changed from12.2 $\mu g\ m^{-3}$ to 33.4 $\mu g\ m^{-3}$ and 60% to 46%, respectively.
We also investigated the relationship between relative humidity (RH) and volume
fractions of ammonium sulfate, ammonium nitrate and organic aerosols as shown in Figure 6.
The results indicated that ammonium nitrate increased most significantly with elevated RH. On
the contrary, ammonium sulfate, as another inorganic compound, showed only a moderate
positive correlation with RH and a decrease in the volume fraction was observed in RH values
larger than 75%. This might be associated with liquid phase oxidation of $SO_2$ under high RH
condition, to sulfate aerosol. Increasing RH may provide more atmospheric oxidants and
reaction media for the aqueous-phase oxidation (Zhang et al., 2015). The volume fraction of
organic aerosol showed negative correlation with increasing RH, as presented in Figure 6 (c),
which was maybe due to a faster increasing volume fraction of inorganic aerosol than organic
aerosol.
This direct observation shows that the contribution of inorganic components increased
during this campaign. It is plausible that the increased concentration of $SO_2$ and $NO_2$
are highly associated with this giving rise to the long-term trends observed in Figure 2
(Pan et al., 2016; Wang et al., 2014).
**4. Conclusion and implication for atmospheric air pollution control**
Atmospheric pollution and associated haze events has a dramatic effect on climate
change, human health and visibility degradation (Ding et al., 2013; Petäjä et al., 2016;
Wang et al., 2015; Zhang et al., 2015). Here, longterm visibility measurements
combined with satellite data sets, in-situ measurements and model results revealed that
increased fractions of inorganic aerosol components in the particle matter are crucial in
contributing to more haze events from 1980-2010. In this way, aerosol hygroscopic
growth has occurred in lower relative humidity conditions than before due to more
ammonium nitrate aerosol, and the light scattering ability of atmospheric aerosol
enhanced as shown in Figure 7. Another mechanism is that high concentration of $NO_x$
can promote the conversion of $SO_2$ to form sulfate aerosol via aqueous phase oxidation
during intensive pollution periods (He et al., 2014; Wang et al., 2016). Considering the
vast energy consumption in the future decades and the sources of inorganic components
in atmospheric aerosol, we demonstrate that the reduction nitrate, sulfate, ammonium
and their precursors should be continued to get better visibility in China.

Acknowledgements
We acknowledge Dr B. Boys and Professor R. Martin of Dalhousie University for providing
GEOS-Chem model results in China. We acknowledge the free use of tropospheric $NO_2$ and $SO_2$
column data from the SCIAMACHY sensor from www.temis.nl. This work was supported by the
Ministry of Science and Technology of China (No: 2017YFC0210000), the National Research
Program for key issues in air pollution control (DQGG0101), the National Natural Science
Foundation of China No.41775162 and Academy of Finland via Center of Excellence in
Atmospheric Sciences and the National Natural Science Foundation of China (41605119).

**Competing financial interests**
The authors declare no competing financial interests.
**Author contributions**
Y.H.W had the original idea. L.L.W and C.S.G provided and processed satellite and
visibility data. Y.S.W provided measurements of  aerosol chemical composition
data.Y.H.W, Y.S.W, L.L.W, T.P and M.K  interpreted the data and write the paper.
All the authors commented on the paper.

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

Figure caption

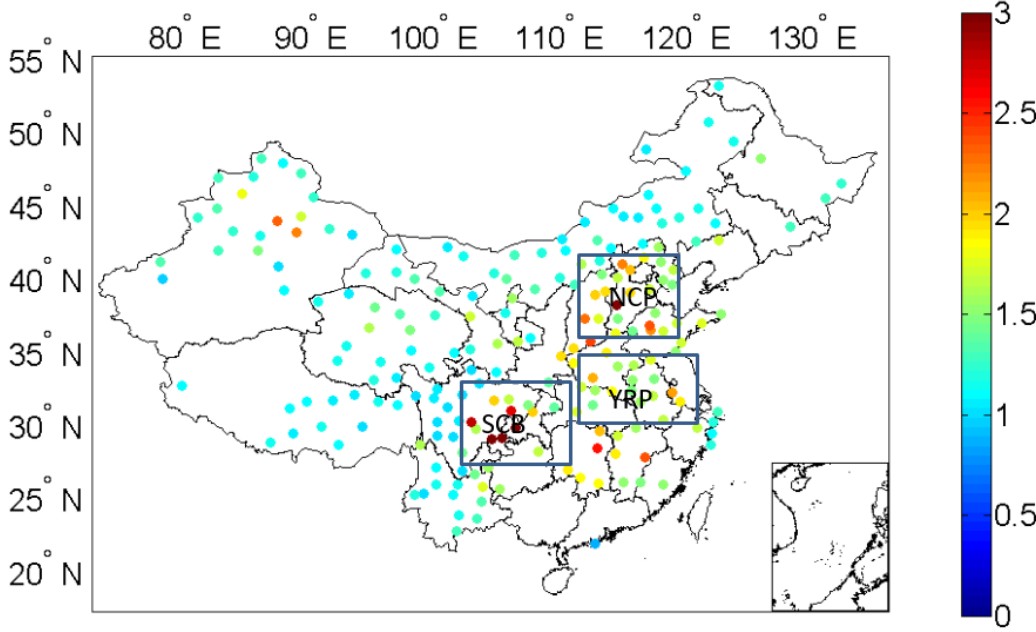


**Figure 1.** The distribution of the average surface visibility ratio in dry and wet

conditions based on observations at 262 surface observation sites in China. The

aerosol in the industrialized regions of China in the East are more hygroscopic than

aerosol particles in the west of China.









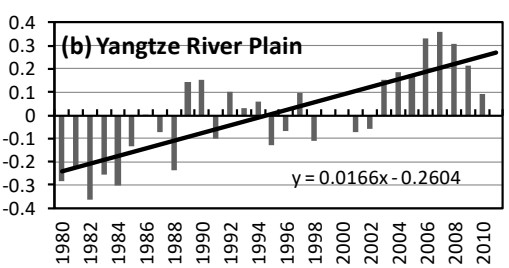


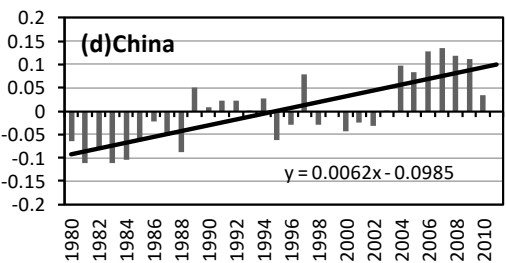


**Figure 2.** Anomalies and trends of ratio of visibility in North China Plain, Yangtze

Plain, Sichuan Basin and in China as a whole. The relative contribution of

hygroscopic aerosols to the visibility reduction has increased from 1980 to 2010 in

China.









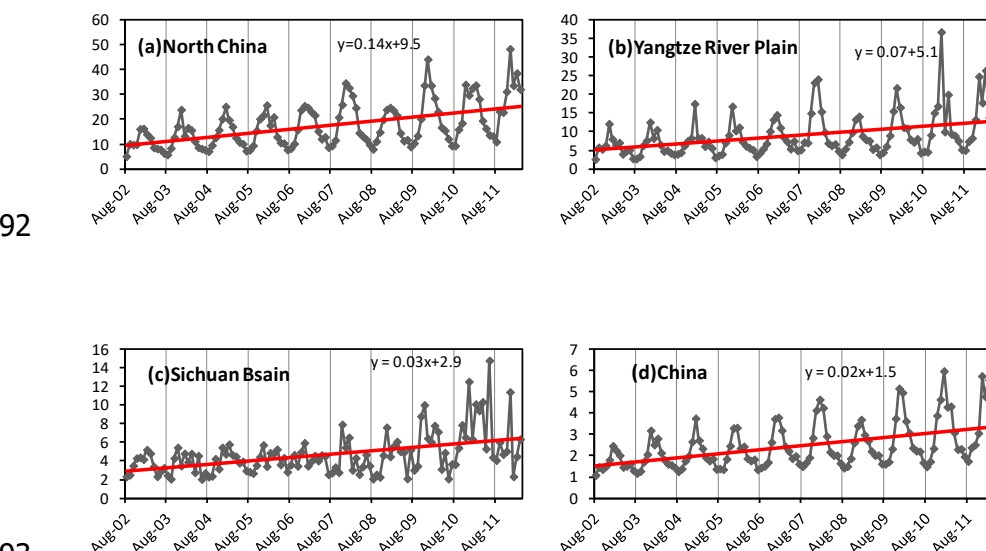



Figure 3. Trends of $NO_2$ concentration over china from SCIAMACHY from the year

2002 to 2012 ($10^{15}$ mol/cm$^2$)











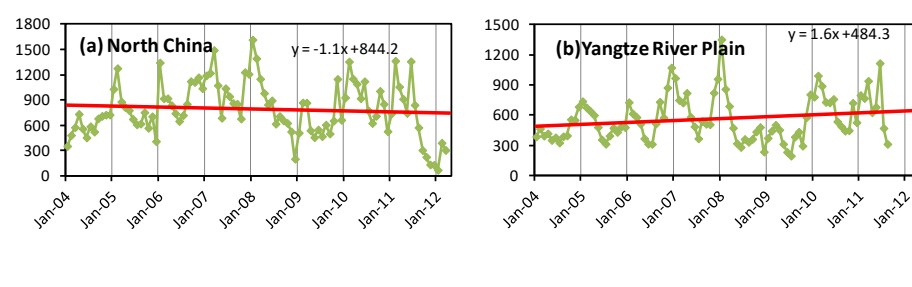

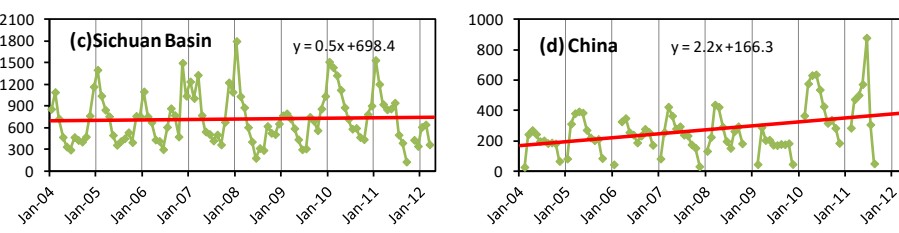

Figure 4. Trends of $SO_2$ concentration over china from SCIAMACHY from the year of

2004 to 2012 (1000DU)









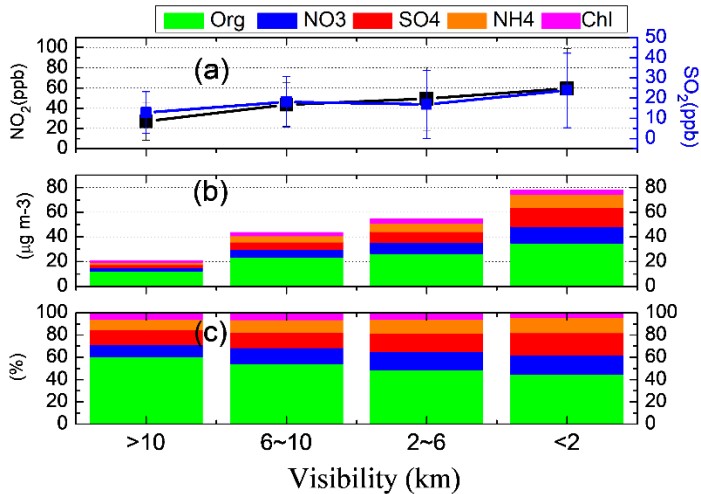

**Figure 5:** Variation of (a) $NO_2$, $SO_2$, (b) chemical composition (c) mass fraction of

organic, nitrate, sulfate, ammonium and chloride with decreased visibility during the

intensive campaign in Beijing.

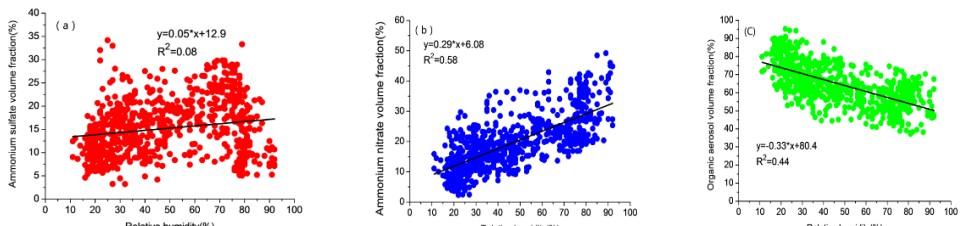


Figure 6. Relation between relative humidity (RH) and volume fractions of (a) ammonium
sulfate (b) ammonium nitrate (c) organic aerosol.












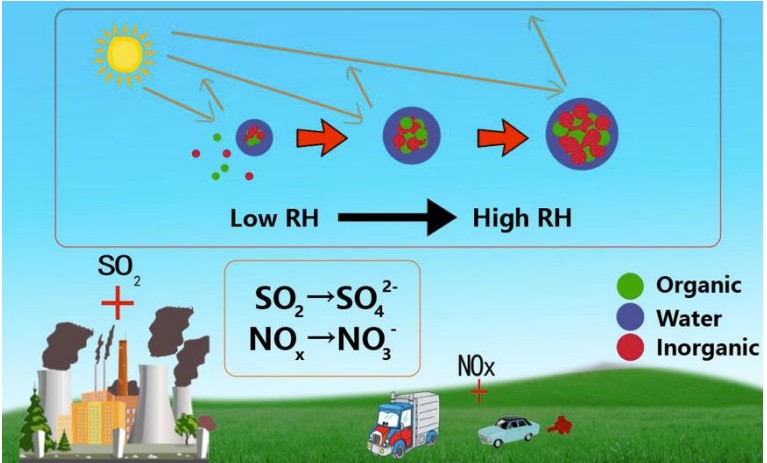
Figure 7. A schematic picture illustrating the process of enhanced emission of
aerosol inorganic precusors and formation of aerosol inorganic components leading to
increased hygroscopicity and aerosol water uptake ability leading to considerable
visibility degradation in China. The plus symbols represents the strengthening of a
specific process.
