# Peer review of "Increased inorganic aerosol fraction contributes to air pollution and"

_Atmospheric Chemistry and Physics, 2018_

## Referee Comment (RC1) · Anonymous Referee #1 · 13 Nov 2018

This paper analyses long-term trends in visibility across China and explores if there is any evidence for aerosol hygroscopicity contributing during haze events using satellite data and ground level measurements of aerosol composition. Overall, visibility has decreased in during periods of high humidity in areas of Chin that experience high levels of haze. The satellite datasets revealed corresponding long-term trends of increasing concentrations of NO2 and SO2, while measurements from one campaign with an aerosol mass spectrometer showed higher fraction of inorganic species. The authors conclude from these two datasets that an increased fraction of sulphates and nitrates may be contributing to increased haze levels as indicted by the decreasing visibility.

While I commend the authors for producing a sharp and focused paper, I feel at times some necessary and important details were lacking that made it hard to understand

the methods and the how the different conclusions were reached (for details see the specific comments below). In my opinion, a bit more in-depth analysis is needed in to link the different sections/datasets. For example, does the observed increase inorganic aerosol fraction correspond to what was observed by the satellite data for that period? Finally, I was not sure what the new or novel findings in this paper compared to previous works in the literature as there have been a number of papers showing the role of inorganic aerosols in haze events (including papers cited by the authors, such as Pan et al., 2016). Perhaps the authors could more clearly state what is new. The paper is reasonably well written and the figures clear and appropriate.

Specific comments

1. The last paragraph of the introduction, in my opinion is normally the aims and objectives of the paper not a summary of the conclusions. 2. Page 4, line 80: The authors state that they use the ratio, Ri, to infer the long term trends in aerosol hygroscopcity but do not mention how or why this ratio gives this information. I think more information is needed. 3. Page 5, line 103. What time series? Fig 1 is a map of average visibility in different locations. Please refer to the reader to actual figure. 4. Page 5, line 108: The authors state that the "enhancement factor due to hygroscopicity is within expected values" yet I do not see where these enhancement factors numbers are given (which figure/Table?) or how they are calculated from the Ri? please give some more information as I cant see how you can 5. Paragraph starting page 6, line 124: In this paragraph, the authors compared their results with other work. To help the reader, it would help I think to state what techniques the different papers used (i.e. modelling, satellite data?) as this will affect 6. Page 7, line 149: Not sure what you mean by a constant trend? 7. Page 7 Line 153-4: What do you mean by the aerosol precursor to NO2? And was the study by Xing et al also over the same period as this work? And was the data from Xing et al also satellite data? 8. Page 7, line 158 and 162: Can you really call NO2 and SO2 satellite data as 'surface observations' or 'ground-based observations'? Or are you referring to something else? 9. Page 7, line 159: what modelling results are you referring to? Please give more details on the actual findings and outputs from the model. For example what time period are you referring to? And how much did it increase by? 10. Section 3.3: I realise that you may not have had access to in-situ measurements of aerosol composition from other locations but I still think you need to comment on how representative Beijing is for the rest of China 11. Page 8, line 174: These are not really big increases in % mass fractions (2%-7%). Firstly, are these statistically significant differences between low and high visibility? And if so, are these small increases really going to have a large impact on aerosol hygroscopicity and therefore visibility? The nitrate fraction only increased by 2%. What was the RH doing during these measurements? Can you based on the composition data calculate the change in aerosol hygroscopicity from low to high visibility? 12. Page 8, Line 177: Does the satellite data also show an increase in NO2 and SO2 during this period (as these are precursors)? 13. Figure 1: need to label the key 14. Figures 2,3,4: need to label the y axis 15. Figure 5: need to label the x axis. Also are these the mean values plotted for each visibility bin? If so I think you should give a indication of the variability (I.e. standard deviation). 16. Figure 6: Not sure what this adds to paper, I didn't quite see how the findings from this paper added to current understanding?

---

## Referee Comment (RC2) · Anonymous Referee #2 · 30 Nov 2018

The authors analyzed the trends of visibility using the dataset of 262 surface observation sites in China, the trends of SO2 and NO2 using the satellite data. They also conducted an intensive campaign during the winter of 2010, and drew a conclusion that increased inorganic fractions in the aerosol particles are the key component in haze events. In general, this study do provide useful information. But I have several key comments that need to be addressed before can be published in ACP.

1. Increase of the inorganic species control the aerosol pollution episodes in east China is not new. Plenty previous studies have provide similar results. In addition, an intensive campaign but not a long term measurement dataset was deployed here to reach the above-mentioned conclusion. This make the conclusion not reliable enough. For example, is there some cases that the PM2.5 increase was dominated by organics?

[Figure]

More deeply and detailed analysis is needed.

2. Since strong control measures were conducted by Chinese government from around 2007-2010, the trends analysis here, which was stopped at 2010, is a bit tricky. The trend after 2010 would be more interesting.

3. The trends showed in Figure 2 was in recent years e.g. after 2007, was decrease but not increase. This should be correlated to the emission reduction I mentioned above.

4. In Figure 5, the increase of organics was large although its fraction decreased. Even in the lowest visibility bin, the contribution of organics was comparable to inorganic speices.
* * *

---

## Author Comment (AC1) · 7 Mar 2019

**A point to point response to the reviewers' comments**

We thank the two reviewers for their comments, and we do think their comments and suggestions improved our manuscript. Here are points to points responses (in blue colored), accordingly, we also revised manuscript (in red colored).

1#Reviewer:

This paper analyses long-term trends in visibility across China and explores if there is any evidence for aerosol hygroscopicity contributing during haze events using satellite data and ground level measurements of aerosol composition. Overall, visibility has decreased in during periods of high humidity in areas of China that experience high levels of haze. The satellite datasets revealed corresponding long-term trends of increasing concentrations of NO2 and SO2, while measurements from one campaign with an aerosol mass spectrometer showed higher fraction of inorganic species. The authors conclude from these two datasets that an increased fraction of sulphates and nitrates may be contributing to increased haze levels as indicted by the decreasing visibility. While I commend the authors for producing a sharp and focused paper, I feel at times some necessary and important details were lacking that made it hard to understand the methods and the how the different conclusions were reached (for details see the specific comments below). In my opinion, a bit more in-depth analysis is needed in to link the different sections/datasets. For example, does the observed increase inorganic aerosol fraction correspond to what was observed by the satellite data for that period? Finally, I was not sure what the new or novel findings in this paper compared to previous works in the literature as there have been a number of papers showing the role of inorganic aerosols in haze events (including papers cited by the authors, such as Pan et al., 2016). Perhaps the authors could more clearly state what is new. The paper is reasonably well written and the figures clear and appropriate.

We thank the reviewer for the thoughtful comments. The main comments could be summariesd as the two aspects:

1.  The method part is hard to understand due to some necessary and important details were lacking.

    Response: Thanks for the comment. Atmospheric light extinction consists from light scattering and light absorption. Among them, light scattering dominates total light extinction. Aerosol light scattering hygroscopic growth factor $f$(RH) is defined as scattering coefficient at low RH with that at high RH and it varies a lot with RH as we shown in figure a. A higher $f$ (RH)

value usually corresponds to higher RH and higher inorganic aerosol fraction, while a lower value usually corresponds to lower RH and high organic fraction. The reason is that inorganic aerosol compounds of nitrate, sulfate and ammonium have more strong water uptake ability than organic compounds (see table S 1). In addition, the high humidity condition in ambient also prefers the formation of inorganic aerosol from precursors of $NO_2$ and $SO_2$. In this study, for a given site and given year, we defined a $f$ (RH)-like parameter, Ri, using the observed annual visibility (V) as a ratio ($R_i$) using visibility values from the surface observation stations We added more explanations on the method part and we hope these could make the study easy to follow. 'Aerosol size growth and composition change in high humidity condition are highly related light scattering ability (Zhang et al., 2015). Studies always use $f$ (RH), a parameter which is defined as the ratio of light scattering coefficient under high RH with that under low RH. $f$ (RH) is a unitless number, usually ranges from one to two. At ambient RH around 80%, a higher $f$ (RH) value usually corresponds to higher inorganic aerosol fraction, while a lower value usually corresponds to high organic fraction. The reason is that inorganic aerosol compounds of nitrate, sulfate and ammonium have more strong water uptake ability than organic compounds (see SI table 1). In addition, the high humidity condition in ambient prefers the formation of inorganic aerosol from precursors of $NO_2$ and $SO_2$ (Wang et al., 2014). In this study, for a given site and given year, we defined a $f$ (RH)-like parameter, Ri, using the observed annual visibility (V) as a ratio ($R_i$) between visibility values from the surface observation stations, when the daily average RH was below 40% for more than 20 days.'

[Figure]

Figure a. Aerosol light scattering hygroscopic growth factor as a function of RH during January and February of 2013 in Beijing. The scattering coefficient at 550nm was measured by nephelometer (Wang et al., 2015).

2.  The novelty of the study should be addressed compared with previous reports.

Response: Thanks for the comment. Compared with previous studies, such as Pan et al., 2016, this study focused on large spatial and temporal scale to illustrate the process. In previous studies, they only used in-situ measurement data in one site for couple of months. In this study, we used the visibility and RH data in eastern of China from 1980-2010, and got a hypotheses that increased fraction of inorganic aerosol is a reason that lead to frequent occurred haze episodes, then we used data from AMS and NO2 and SO2 to validate the hypotheses. Moreover, we added more data, figures and explanations in our revised version. In our opinion, the study could confirm that emission of inorganic precursors, like SO2 and NO2 are more increased than organic precursors as we shown Figure B as following, which lead to more frequent occurred hazes in China.

[Figure]

[Figure]

[Figure]

Figure B Yearly averaged concentrations of (1) NO2(2) SO2 (C ) CH2O in China , all three data are derived from SCIAMACHY satellite. The concentration of CH2O could be used as a proxy of VOCs.

Specific comments

1.  The last paragraph of the introduction, in my opinion is normally the aims and objectives of the paper not a summary of the conclusions.

    Response: We agree, the last paragraph has been revised as following:

    Here, a comprehensive data sets were used to reveal that an increasing trend of inorganic components in atmospheric aerosol may be a pivotal factor, at least, which leads to frequently occurred haze events in China from 1980-2010. We suggest that the controlling of inorganic aerosol components of nitrate, sulfate and their precursors should be of a high priority due to their strong water uptake abilities and therefore, light scattering ability in high RH conditions.

2.  Page 4, line 80: The authors state that they use the ratio, Ri, to infer the long term trends in aerosol hygroscopicity but do not mention how or why this ratio gives this information. I think more information is needed.

    Response: Thanks for the comments. Please see more explanations and statement to first general comment.

3.  Page 5, line 103. What time series? Fig 1 is a map of average visibility in different locations. Please refer to the reader to actual figure.

    Response: Thanks for the comments. The time series is 1980-2010 and we added deeper analysis on this part as following: The average visibility in low RH in NCP, SCB, YRP and China

are 18.2 km, 21.4 km, 19.5 km and 23.3 km, while in high RH conditions are10.6 km, 13.7 km, 13.7 km and 17.4 km, respectively. In general, visibility in low RH condition has fluctuated trend, particularly in YP and SB region, whereas visibility in high RH conditions showed decreasing trend as we shown in Figure S1 (a) and (b) . The average ratio of visibility in low RH to that in high RH from 1980-2010 is presented in Figure 1. The maximum ratios were identified in eastern China and in some western Chinese cities. Three heavily polluted regions, Northern China Plain, Sichuan Basin and Yangtze river Plain were identified based on values of $R_i$, which are also constant with aerosol mass concentrations and haze day distributions    (van Donkelaar et al., 2010; Xin et al., 2015).

The average $R_i$ during 1980-1984 in Northern China Plain, Sichuan Basin and Yangtze river Plain are 1.62, 1.41, 1.29 and 1.31, respectively, contrasting with the values of 1.98, 1.81, 1.70 and 1.52 during 2006-2010. The increments are 22.3%, 27.3%, 31% and 16%, respectively. It is worth noting that the $R_i$ in Yangtze river Plain region exhibits the most increment, which implies the increased emissions with rapid economic growth.

4. Page 5, line 108: The authors state that the "enhancement factor due to hygroscopicity is within expected values" yet I do not see where these enhancement factors numbers are given (which figure/Table?) or how they are calculated from the Ri? please give some more information as I cant see how you can

Response: Thanks for the comment, what we wanted to express was that high Ri values in these regions indicated that these regions suffered severe pollution as suggested by mode result and observation result. We changed statement as following: Three heavily polluted regions, Northern China Plain, Sichuan Basin and Yangtze river Plain were identified based on high values of $R_i$, which were also constant with aerosol mass concentrations and haze day distributions    (van Donkelaar et al., 2010; Xin et al., 2015)

5. Paragraph starting page 6, line 124: In this paragraph, the authors compared their results with other work. To help the reader, it would help I think to state what techniques the different papers used (i.e. modelling, satellite data?) as this will affect

Response: Thanks for the comment, we added the techniques that they used as in our

revised manuscript.

6. Page 7, line 149: Not sure what you mean by a constant trend?

Response: Thanks for the comment. We have changed the statement as' but increased mostly in China from 2004 to 2012'.

7. Page 7 Line 153-4: What do you mean by the aerosol precursor to NO2? And was the study by Xing et al also over the same period as this work? And was the data from Xing et al also satellite data?

Response: Thanks for the comment. It should be that NO2 is an aerosol precursor. And the study by Xing et al was a modeling work from 1990-2010, which have overlap of NO2 measurement from 2002-2010. We changed the statement as: Anyway, as an important aerosol precursor, $NO_2$ showed the most increasing trend in China, consistent with the trend of increased aerosol concentration by modeling result (Xing et al., 2015) .

8. Page 7, line 158 and 162: Can you really call NO2 and SO2 satellite data as 'surface observations' or 'ground-based observations'? Or are you referring to something else?

Response: Thanks for the comment. We can not call NO2 and SO2 satellite data as 'surface observation' or 'ground-based observations'. Here we are referring that modeled aerosol compounds are consistent with observations work.

9. Page 7, line 159: what modelling results are you referring to? Please give more details on the actual findings and outputs from the model. For example what time period are you referring to? And how much did it increase by?

Response: Thanks for the comment. The results are from modeling combined with satellite output from 1998-2012, which is referring to Boy et al., (2014). Their results clearly showed increased PM2.5 mass concentration from 1998-2012 with 0.79 $\mu g\ m^{-3}$ per year in east Asian. We also used their result and showed an increased trend of inorganic aerosol fraction in Figure s4.

10. Section 3.3: I realize that you may not have had access to in-situ measurements of aerosol composition from other locations but I still think you need to comment on how representative Beijing is for the rest of China.

Response: Thanks for the comment. Since we can not get aerosol composition data from other locations, we used Beijing as an example, we not claimed that Beijing could be exactly as the

representative of the rest of China, although the position of Beijing is North China Plain. The climate of the Beijing is typicality driven by east Asian monsoon, which is the most important driven factor for the climate.

11. Page 8, line 174: These are not really big increases in % mass fractions (2%-7%). Firstly, are these statistically significant differences between low and high visibility? And if so, are these small increases really going to have a large impact on aerosol hygroscopicity and therefore visibility? The nitrate fraction only increased by 2%. What was the RH doing during these measurements? Can you based on the composition data calculate the change in aerosol hygroscopicity from low to high visibility?

Response: Thanks for the comprehensive comment. In Figure 5 (a), we used NO2 instead of NOx in our revised manuscript since NO2 is more relevant with nitrate aerosol. We divided your comment as the four separate comments: a; These are not really big increases in % mass fractions (2%-7%). Firstly, are these statistically significant differences between low and high visibility? We need to apologies that the statistic was not correct in the previous version, the mass fraction of these inorganics should increase from 11.3% to 17.3%, from 13.0% to 19.9% and from 9.6% to 13.6%, respectively. The total increment was around 17%, and the statistically significant differences. b: And if so, are these small increases really going to have a large impact on aerosol hygroscopicity and therefore visibility? Yes, according to water uptake ability of inorganic and organic compounds as we listed in Table S1, the kappa value are 5-6 times higher for inorganic than organic. So, the difference will lead to more aerosol liquid water content due to inorganic aerosol, thereby, light scattering enhancement during pollution period. c:The nitrate fraction only increased by 2%. What was the RH doing during these measurements? The nitrate fraction increased from 11.3 % to 17.3% from our revised statistics. The effect of RH will lead to aerosol water uptake, especially for inorganic compounds. Furthermore, increased water content in aerosol could lead to more efficient light scattering as we shown in Figure A. d: Can you based on the composition data calculate the change in aerosol hygroscopicity from low to high visibility?    From low visibility to high visibility usually corresponded to high pollution period to clean period, we cannot calculate exactly aerosol light hygroscopic growth from low to high visibility/ high visibility to low visibility Since we did not have size resolved aerosol chemical composition data from 100 nm to 1000 nm. However, for a specific site like Beijing, we could give

a parameterized value according Figure A, which is obtained during January of 2013. The *f*(RH) could doubled from low RH to high RH condition.

12. Page 8, Line 177: Does the satellite data also show an increase in NO2 and SO2 during this period (as these are precursors)?

Response: Thanks for the comment. We plotted column concentrations of NO2 and SO2 from September 2010 until April of 2011. The campaign period was November 2010 to February of 2011. From Figure C we can see that the concentration of NO2 was much high during campaign period, while the SO2 did not shown always high concentrations.

[Figure]

Figure C. Monthly column concentrations of NO2 and SO2 in NCP region from satellite observation.

13. Figure 1: need to label the key

Response: Thanks, the unit of y axis is unitless as we explained in the method part.

14. Figures 2,3,4: need to label the y axis

Response: Thanks for the suggestion. We labeled the y axis in our figure caption part, since the space for the y axis is not enough to label clearly.

15. Figure 5: need to label the x axis. Also are these the mean values plotted for each visibility bin? If so I think you should give an indication of the variability (I.e. standard deviation).

Response: Thanks for the comment, We labeled the x axis. These are the mean values and we have added standard deviation for each visibility bin for NO2 and SO2, as shown in the figure according to your comments.

16. Figure 6: Not sure what this adds to paper, I didn't quite see how the findings from this paper added to current understanding?

Response: Thanks for the comment. We give a schematic picture to illustrate the process

of enhanced emission of aerosol inorganic precursors and formation of aerosol inorganic components leading to increased hygroscopicity and aerosol water uptake ability leading to considerable visibility degradation in eastern China from 1980-2010. In our opinion, our study is more focused on large spatial and temporal scale to illustrate the process.

Reviewer #2

The authors analyzed the trends of visibility using the dataset of 262 surface observation sites in China, the trends of SO2 and NO2 using the satellite data. They also conducted an intensive campaign during the winter of 2010, and drew a conclusion that increased inorganic fractions in the aerosol particles are the key component in haze events. In general, this study do provide useful information. But I have several key comments that need to be addressed before can be published in ACP.

Response: We thank the reviewer for the positive comments.

1.  Increase of the inorganic species control the aerosol pollution episodes in east China is not new. Plenty previous studies have provide similar results. In addition, an intensive campaign but not a long term measurement dataset was deployed here to reach the above-mentioned conclusion. This make the conclusion not reliable enough. For example, is there some cases that the PM2.5 increase was dominated by organics? More deeply and detailed analysis is needed.

    Response: Thanks for the comment. In the revised version, we added more figures, statements and explanations. We hope these revisions will make the manuscript improved. After we done the statistics of our AMS and filter data, we found that during clean sky conditions, the concentration of NR-PM1 (from AMS) and PM2.5 is dominated by organic matter (around 60%~80% mass fraction). However, during haze period, the concentration of inorganic increased much more than organic, despite the mass concentration of inorganic and organic increased. So, during the haze period, the increased of PM2.5 is always dominated by inorganic.

2.  Since strong control measures were conducted by Chinese government from around 2007-2010, the trends analysis here, which was stopped at 2010, is a bit tricky. The trend after 2010 would

be more interesting.

Response: Thanks for the comment. We agree with that strong control measures were conducted by Chinese government from around 2007-2010. However, satellite observed column concentrations of NO2 and SO2 still show increased trends in Northern China Plain, Sichuan Basin and Yangtze river Plain as we shown in manuscript figure 3 and 4 in revised manuscript, The real emission reduction was from the beginning of 2013, the central government of China took lots of measures to improve air quality in Beijing-Tianjin-Hebei (BTH), Yangtze River delta (YRD) region and Pearl River delta (PRD) region. In particular, the state council released clean air action in September of 2013, called clean air action, aiming to reduce concentrations of $PM_{2.5}$ in BTH, YRD and PRD in the next five years. Actually, we have a paper just accepted by Science China: Earth Science, which has comprehensively evaluated trends of aerosol mass concentration, precursors, compositions in Beijing-Tianjin-Hebei (BTH), Yangtze River delta (YRD) region and Pearl River delta (PRD) region from 2013-2017. The results confirmed that reduction of aerosol mass concentration in three regions, and the reduction is mainly due to decreased concentrations of nitrate and sulfate.

3. The trends showed in Figure 2 was in recent years e.g. after 2007, was decrease but not increase. This should be correlated to the emission reduction I mentioned above.

Response: Thanks for the comment. We think you are referring figure 3 and figure 4, from which we can see decreases of NO2 and SO2 after 2007. We have explanations in our manuscript.' A decreasing trend was observed during the year of 2008 and 2009, especially in Northern China Plain. This may be due to a combination of Chinese economic downturn and emission reduction during the Olympic games (Lin and McElroy, 2011) (Wang et al., 2010)'.

4. In Figure 5, the increase of organics was large although its fraction decreased. Even in the lowest visibility bin, the contribution of organics was comparable to inorganic species.

Response: Thanks for the comment. We agree that the mass concentration of inorganic and organic increased significant from high visibility to low visibility. However, consider the different water uptake ability of inorganic compared with organic (kappa), as we shown in the table below, The kappa values for NH4NO3 and (NH4)2SO4 are 0.68 and 0.53, respectively, while kappa is only 0.1 for organic

compounds, so we believe that the difference will lead to more aerosol liquid water content due to inorganic aerosol, thereby, light scattering enhancement during pollution period.

Table A Hygroscopic growth factors kappa (κ) for pure substance

| Substance | κ (at $\alpha\omega$ =0.85) |
|---|---|
| $NH_4NO_3$ | 0.68 |
| $(NH_4)_2SO_4$ | 0.53 |
| $NH_4HSO_4$ | 0.56 |
| $H_2SO_4$ | 0.97 |
| Organic | 0.1 |

Reference

Boys, B.L. et al. Fifteen-Year Global Time Series of Satellite-Derived Fine Particulate Matter. Environmental Science & Technology, 48(19): 11109-11118, 2014.

Pan, Y. et al. Redefining the importance of nitrate during haze pollution to help optimize an emission control strategy. Atmospheric Environment, 141: 197-202, 2016.

Wang, Y.H. et al. Aerosol physicochemical properties and implications for visibility during an intense haze episode during winter in Beijing. Atmos. Chem. Phys., 15(6): 3205-3215, 2015.

Xing, J. et al. Observations and modeling of air quality trends over 1990–2010 across the Northern Hemisphere: China, the United States and Europe. Atmos. Chem. Phys., 15(5): 2723-2747, 2015.

---

## Author Response (AR2)

Dear editor,

Thank you very much for your careful reading and comments on our manuscript. We appreciated that your suggestions and comments have improved our paper. Accordingly, we responded your comments point by point, as shown bellowing.

Abstract

Please state the time period analyzed by the study. This is important, given the rapid changes in emissions and pollutants in China.

Response: We added time period in our revised version.

This last statement in the abstract maybe needs some thought: "and particularly the reduction of nitrate, sulfate and their precursor gases would contribute towards better air quality in China". Reduction in these components would improve visibility due to aerosol water uptake as you have demonstrated, but is this the same as improving air quality?

This statement also needs to include the caveat that SO2 and NOx emissions have already declined substantially since the period studied in this paper.

Response: We revised the statement as' particularly the reduction of nitrate, sulfate and their precursor gases would contribute towards better visibility in China. '

Introduction

A concise review of previous work on trends in air pollution over China in recent years would be helpful. This was suggested by Referee #2, but not fully addressed in the revised manuscript. A paper that I co-authored (Silver et al., 2018) includes relevant references that could be cited here:

https://iopscience.iop.org/article/10.1088/1748-9326/aae718/meta

Response: We added    previous work on the air pollution in recently years, as' A recent study have suggested significantly    decreased trends of $PM_{2.5}$ and $SO_2$ in China from 2015-2017 by analyzing data sets from Ministry of Ecology and Environment of China

(Silver et al., 2018). The column $NO_2$ concentration obtained from OMI showed increased trend during 2005-2011, while a decreasing trend during 2012-2015 (Itahashi et al., 2016). The $SO_2$ concentration has decreased around 50% from 2012-2015 in North China Plain due to economic slowdown and governments efforts to restrain emissions from power and industrial sectors (Krotkov et al., 2016).'

Methodology

More additional details on the methods are needed to help the reader understand the paper. This was suggested by Referee #2 and not fully addressed in the revised manuscript. Specifically:

1) Time periods of the different datasets, particularly the visibility dataset.

2) Please provide brief details on the model product that you use. You use a model-satellite derived product and this needs to be stated.

3) More details on the AMS data. In particular, time period and location.

4) Details on the SCIAMACHY product including a reference.

Response: Please see revised version, we revised corresponding places as you suggested.

Section 3.1 There is an indication in Figure 2 that the long-term trend reversed in around 2007 to 2008, approximately the same time previous studies have reported a peak in SO2 and NOx emissions in China. Given the previous literature on this point I think this requires some brief discussion. See the related comment from Referee #2 on this point.

Response: Thanks for the comment, Actually, the concentrations of SO2 and NO2 decreased around 2008, also from previous work (please see figure as following) (Krotkov et al., 2016). The decreases in our work are consistent with their work. The decreases were due to combination of Chinese

economic downturn and emission reduction during the Olympic games.

[Figure]

Section 3.2 See Referee #2 comment on trends. There have been a number of papers written on NO2, SO2, and

PM2.5 trends over China that are not referred to in the revised manuscript. It is not clear how the revised manuscript

either confirms or contradicts previous analysis. Please clarify. Some important papers that report recent trends in

SO2 and NO2 over China include:

Response: Please see revised version, we added this in our revised version

Krotkov N A et al 2016 Aura OMI observations of regional SO2 and NO2 pollution changes from 2005 to 2015

Atmos. Chem. Phys. 16 4605–29

Ling Z, Huang T, Zhao Y, Li J, Zhang X, Wang J, Lian L, Mao X, Gao H and Ma J 2017 OMI-measured increasing

SO2 emissions due to energy industry expansion and relocation in northwestern China Atmos. Chem. Phys. 17 9115–

31

Van Der A R J, Mijling B, Ding J, Elissavet Koukouli M, Liu F, Li Q, Mao H and Theys N 2017 Cleaning up the air:

effectiveness of air quality policy for SO2 and NOx emissions in China Atmos. Chem. Phys. 17 1775–89

Irie H, Muto T, Itahashi S, Kurokawa J and Uno I 2016 Turnaround of tropospheric nitrogen dioxide pollution trends

in China, Japan, and South Korea Sola 12170–4

PM2.5 concentrations have also declined recently across China as demonstrated by surface observations (Silver et al., 2018) and satellite studies. Does the increase in NO2 and SO2 after 2007 contradict previous analysis. This requires some comment. Section 4. It is important to note that the present study ends in 2010 (2012?). There have been large reductions in Chinese NOx and SO2 emissions since 2012. This is a crucial caveat that needs to be mentioned in the conclusions.

Response: Thanks for the comment. The decreased trend of NO2, SO2 and PM2.5 were confirmed by some studies. However, we only analyzed visibility data from 1980-2010 and satellite data until 2012, so our conclusion was made very carefully. In our revised version, we added statement on time periods as the editor suggested. Also, we also revised our conclusion as 'Considering the vast energy consumption in the future decades and the sources of inorganic components in atmospheric aerosol, we demonstrate that the reduction nitrate, sulfate, ammonium and their precursors should be continued to get better visibility in China'.

Figure 5. Add an explanation of the x-axis to the figure caption.

Response: We labeled X-axis as visibility in our revised version.